# Socioeconomic position and eye health outcomes: identifying inequality in rapid population-based surveys

Ian McCormick [ID],[1] Min J Kim [ID],[1] Abba Hydara [ID],[2] Segun I Olaniyan [ID],[2] Modou Jobe [ID],[3] Omar Badjie,[4] Nyakassi M B Sanyang,[5] Gibril Jarju,[6] Modou Njai,[4] Alhagie Sankareh,[7] Andrew Bastawrous [ID],[1] Luke Allen [ID],[1] Islay Mactaggart [ID],[1] Matthew J Burton [ID],[1,8] Jacqueline Ramke [ID] [1,9]

For numbered affiliations see end of article.

**Correspondence to**
Mr Ian McCormick;
ian.mccormick@lshtm.ac.uk

## ABSTRACT

**Objective** Monitoring health outcomes disaggregated by socioeconomic position (SEP) is crucial to ensure no one is left behind in efforts to achieve universal health coverage. In eye health planning, rapid population surveys are most commonly implemented; these need an SEP measure that is feasible to collect within the constraints of a streamlined examination protocol. We aimed to assess whether each of four SEP measures identified inequality—an underserved group or socioeconomic gradient—in key eye health outcomes.

**Design** Population-based cross-sectional survey.

**Participants** A subset of 4020 adults 50 years and older from a nationally representative sample of 9188 adults aged 35 years and older in The Gambia.

**Outcome measures** Blindness (presenting visual acuity (PVA) <3/60), any vision impairment (VI) (PVA <6/12), cataract surgical coverage (CSC) and effective cataract surgical coverage (eCSC) at two operable cataract thresholds (<6/12 and <6/60) analysed by one objective asset-based measure (EquityTool) and three subjective measures of relative SEP (a self-reported economic ladder question and self-reported household food adequacy and income sufficiency).

**Results** Subjective household food adequacy and income sufficiency demonstrated a socioeconomic gradient (queuing pattern) in point estimates of any VI and CSC and eCSC at both operable cataract thresholds. Any VI, CSC <6/60 and eCSC <6/60 were worse among people who reported inadequate household food compared with those with just adequate food. Any VI and CSC <6/60 were worse among people who reported not enough household income compared with those with just enough income. Neither the subjective economic ladder question nor the objective asset-wealth measure demonstrated any socioeconomic gradient or pattern of inequality in any of the eye health outcomes.

**Conclusion** We recommend pilot-testing self-reported food adequacy and income sufficiency as SEP variables in vision and eye health surveys in other locations, including assessing the acceptability, reliability and repeatability of each question.

## INTRODUCTION

The WHO and the *Lancet Global Health* Commission on Global Eye Health have

## STRENGTHS AND LIMITATIONS OF THIS STUDY

⇒ Strengths included the use of a nationally representative sample of adults in a country typical of the low-income and middle-income settings surveyed with rapid eye health survey methods.
⇒ Further, the outcomes we report are key eye health indicators included in WHO Eye Care Indicator Menu; effective cataract surgical coverage is one of two World Health Assembly endorsed indicators with a global target for 2030.
⇒ Also, we report both objective and subjective indicators of socioeconomic position (SEP) to enable comparison.
⇒ Limitations include that the most recent objective asset wealth tool at the time of data collection was based on data collected 6 years prior (2013 Demographic and Health Survey).
⇒ In addition, our sample size was insufficient for analysis of intersectionality of SEP with other key equity dimensions.

called for vision and eye health data to be disaggregated by equity dimensions.[1 2] This disaggregation is essential to understand eye health inequalities between population subgroups and to monitor progress towards eye health within Universal Health Coverage. Both WHO and the Commission identified socioeconomic position (SEP) as a key factor in eye health outcomes and the Commission recommended standard SEP indices be included in population-based vision impairment (VI) surveys.[1 2]

Within countries, health inequalities exist along a socioeconomic gradient,[3 4] and SEP indicators identify people experiencing poverty or deprivation via some means of social stratification.[5–7] SEP indicators can be used to measure the differential consequences of the social determinants of health and to monitor and evaluate policies and programmes designed to address inequity.[8–10]

SEP can be described at an individual or group level (eg, household or community) and can be measured in a variety of ways, such as using income, education, occupation, assets, nutrition, housing or a multidimensional index of such factors.[11] Measures can be objective or subjective, and absolute or relative.[11][12] Poverty lines and asset wealth are objective measures, while self-reported, subjective measures of SEP assess individuals' perception of their own resources or standing relative to those living around them.

In comprehensive surveys of vision and eye health, lower SEP has been shown to be associated with higher prevalence of VI and worse access to services,[13–16] though there is variation in SEP measures used. The Rapid Assessment of Avoidable Blindness (RAAB) is the most commonly conducted rapid vision survey worldwide.[17][18] RAAB has streamlined examination protocols compared with comprehensive surveys, but generates key outputs that include the prevalence of VI and effective cataract surgical coverage (eCSC)—both important indicators for monitoring global eye health.[19][20] Currently, age and gender are the only equity dimensions routinely recorded in RAAB while a disability module is optional. To expand data collection to include a measure of SEP within RAAB it is crucial that steps are taken to identify which measures are feasible to collect regularly and might identify those most at risk of being left behind. The anticipated 'time-cost' of incorporating SEP indicators into a rapid survey means more concise questions need to be considered.

In this paper, we aimed to assess whether one objective asset-based measure and three subjective measures of relative SEP demonstrated a socioeconomic gradient or inequality in key eye health outcomes in the population 50 years and older using data from the 2019 Gambia National Eye Health Survey, a comprehensive cross-sectional survey with an embedded RAAB.[21][22] Further, we aimed to assess the level of association between the objective asset-based measure and three subjective measures of relative SEP and recommend one or more indicators to pilot test in other locations.

## METHODS

### Data collection and categorisation

A detailed summary of the 2019 Gambia National Eye Health Survey sampling and examination protocol is provided elsewhere.[21] Briefly, the 2013 National Census was used as a sampling frame to select a nationally representative sample of adults aged 35 years and older in clusters of 30 people.[23] A multi-stage, stratified, cluster random sampling with probability proportionate to size strategy was used. The sample was stratified by historic geographic regions of the country and the urban and rural population proportions within them, using The Gambia Bureau of Statistics' definitions of urban and rural.

At the individual level, sociodemographic data collected included age, sex, marital status, religion, ethnicity, education, occupation and subjective SEP (details below). A questionnaire on asset ownership was completed at the household level via the household head or an adult household key informant (details below). No absolute measure of income or expenditure was collected. Distance visual acuity (VA) was measured indoors at 3 m using the Peek Acuity application on Huawei MediaPad M3 tablet devices[24]; more information is provided in the Eye Health Assessment section of the study protocol.[21] Lens grading was done on slit-lamp examination following dilation with tropicamide 1% eye drops and according to the WHO Cataract Grading Tool.[25] The survey was completed by four data collection teams between February and July 2019.

### Eye health outcomes

We selected WHO priority national eye health indicators as outcomes: VI prevalence and eCSC (table 1).[26] We report blindness and VI according to existing WHO definitions. Cataract surgical coverage (CSC) and eCSC were calculated using the most recent definition.[27] For the purposes of determining service coverage, cataract classified as gradable, mature or hypermature met the definition of *operable*. *Operated* cataract was defined where aphakia (excluding cases of couching) or an intraocular lens was recorded. CSC and eCSC were reported at two best-corrected VA surgical thresholds: <6/12 and <6/60, and the threshold for effective coverage (postoperative presenting VA) was ≥6/12.[19]

### SEP indicators

SEP was assessed in four separate ways. The first was to divide the study population into SEP quintiles using the 'EquityTool' analysis provided by Metrics for Management

| Table 1 | Four key eye health outcomes and their descriptions |
| --- | --- |
| **Eye health outcome** | **Description** |
| Blindness | Presenting VA (ie, with available correction, if worn) in the better eye worse than 3/60 |
| Any vision impairment (VI) | Presenting VA (ie, with available correction, if worn) in the better eye worse than 6/12 |
| Cataract surgical coverage (CSC) | The number of people with operated cataract as a proportion of those having operable plus operated cataract |
| Effective cataract surgical coverage (eCSC) | The number of people with operated cataract and a good visual outcome (presenting VA 6/12 or better) as a proportion of those having operable plus operated cataract |
| VA, visual activity. | |

| Table 2 | Three subjective socioeconomic position variables including the question and possible responses |
|---|---|
| Food adequacy | 'When you think about the food in your household would you say you have (1) Less than adequate food for the needs of your household, (2) Just adequate food for the needs of your household or (3) More than adequate food for the needs of your household'. |
| Income sufficiency | 'When you think about the income in your household would you say it is (1) Not enough to cover our needs, we must borrow, (2) Not enough to cover our needs, we use savings, (3) Just enough to cover our needs, (4) Enough to cover our needs, we are able to save a little or (5) Enough to cover our needs, we are building savings'. |
| Economic ladder question | 'Please look at this ladder (5 step ladder pictured). If the bottom step (1) of the ladder represents the people who are poorest in your community, and the top step (5) represents the people who are richest, on which step of the ladder do you feel your household stands on?' |

(www.m4mgmt.org). EquityTool is an *objective relative SEP measure* that uses a short asset ownership and household characteristics questionnaire to generate wealth quintiles with a minimum level of agreement (kappa ≥0.75) with quintiles derived from underlying (more comprehensive) Demographic and Health Survey (DHS) data. The Equity-Tool for The Gambia used the 2013 DHS and consisted of 16 questions (online supplemental appendix 1).[28] Participants were assigned to a national wealth quintile benchmarked to the 2013 DHS according to their household questionnaire score. In addition to quintile assignment, the EquityTool output includes three continuous variables for national, urban and rural scoring on the questionnaire. We created evenly distributed within-sample quintiles based on the national score for all participants. These were not benchmarked to national thresholds and represented the distribution of asset wealth within the sample only.

For the remaining three questions we used *subjective relative SEP measures* relating to food adequacy, income sufficiency and self-perceived position on an economic ladder (table 2). These questions come from Howe and colleagues' analysis of 11 280 Malawian households that found them to be better at identifying low-income households (those living on less than US$1/day) than a composite wealth index.[8]

## Patterns of inequality
We sought to identify which of the patterns of inequality described in the WHO *Handbook on Health Inequality Monitoring*, were present for our outcomes, namely, mass deprivation, marginal exclusion, queuing or complete coverage.[9] Mass deprivation describes poor health outcomes in all SEP groups except the richest. Marginal exclusion describes worse outcomes in the poorest group compared with all other. Queuing describes an approximately linear decrease in outcomes down SEP groups. Complete coverage describes universally good outcomes across SEP groups.

## Data analysis
In reporting selected eye health outcomes, we restricted our population of interest to people 50 years and older as this age group is directly relevant to the RAAB survey methodology. We assessed the association between the ordinal categories of objective relative asset wealth and each subjective SEP measure by estimating Kendall's tau-b correlation coefficient (assuming a monotonic relationship for all three comparisons). Kendall's tau-b coefficient values range from −1 to 1, with values closer to 1 indicating a stronger relationship and a negative value indicating an inverse relationship.

All analyses were done in Stata V.16. We assigned participants aged 50 years and older to within-sample wealth quintiles using the xtile command. The prevalence of any VI, blindness, CSC and eCSC and their 95% CIs were calculated adjusted for clustering and post-stratified (weighted) according to the age-sex distribution of The Gambia 2013 Census using the svy command. The weighted proportions of participants by sociodemographic and SEP characteristics were calculated according to the age-sex distribution of The Gambia 2013 Census using the svy command.

## Missing data
To prevent listwise deletion during analyses, raw data were checked for completeness. In August 2020, follow-up data collection was completed by telephone to address some missing EquityTool data and following this, 600 forms with persisting missing critical data (one or more entire module (eg, optometry, ophthalmology, mental health), EquityTool data or VA data) were deleted from the final sample.[21] EquityTool questions were imputed for 398 participants (4.3%) in the final dataset (179 (4.5%) participants aged 50 years and older), whereby missing values in a cluster were replaced with the most frequent, non-missing value from within the cluster. Missing values were not substituted for questions that had more than one 'most frequent' response in the cluster (eg, bimodal with equal numbers of observed values for 'yes' and 'no').[21]

## Patient and public involvement
Survey participants were not involved in the design of the survey.

## RESULTS
There were 9188 participants examined out of 11 027 enumerated (response rate 83%). The mean age of the sample was 49.6 years (SD 13.4), 71% were female and 55% were living in an urban location. There were 4020 participants aged 50 years or older included in the

**Table 3** Sociodemographic and socioeconomic position characteristics in the population 50 years and older, The Gambia National Eye Health Survey, 2019

| | Population 50 years and older (N=4020) | | |
|---|---|---|---|
| | N | Crude (%) | Weighted (%) |
| **Sex** | | | |
| Male | 1576 | 39.2 | 50.9 |
| Female | 2444 | 60.8 | 49.1 |
| **Marital status** | | | |
| Married/cohabiting | 2912 | 72.4 | 76.3 |
| Widowed | 1035 | 25.8 | 21.8 |
| Divorced/separated | 57 | 1.4 | 1.4 |
| Never married | 16 | 0.4 | 0.5 |
| **Location** | | | |
| Urban | 2180 | 54.2 | 54.8 |
| Rural | 1840 | 45.8 | 45.2 |
| **Education level** | | | |
| Non-formal (Quranic)* | 2455 | 61.1 | 58.9 |
| Preschool†/no school | 823 | 20.5 | 20.4 |
| Primary† | 229 | 5.7 | 6.1 |
| Secondary†/vocational | 318 | 7.9 | 9.5 |
| Higher | 86 | 2.1 | 2.7 |
| Don't know | 109 | 2.7 | 2.4 |
| **Occupation category** | | | |
| Unemployed | 910 | 22.6 | 21.4 |
| Manual/unskilled | 1987 | 49.4 | 47.8 |
| Trade/skilled manual | 714 | 17.8 | 19.4 |
| Professional | 120 | 3.0 | 3.9 |
| Retired | 238 | 5.9 | 5.9 |
| Other | 51 | 1.3 | 1.6 |
| **National asset wealth quintiles (2013 DHS EquityTool)** | | | |
| Q1—poorest | 366 | 9.1 | 9.4 |
| Q2 | 547 | 13.6 | 14.1 |
| Q3 | 1022 | 25.4 | 25.3 |
| Q4 | 936 | 23.3 | 24.0 |
| Q5—richest | 1149 | 28.6 | 27.3 |
| **Within-sample asset wealth quintiles** | | | |
| Q1—poorest | 814 | 20.3 | – |
| Q2 | 794 | 19.8 | – |
| Q3 | 804 | 20.0 | – |
| Q4 | 804 | 20.0 | – |
| Q5—richest | 804 | 20.0 | – |
| **Food adequacy** | | | |
| Less than adequate | 976 | 24.3 | 24.3 |
| Just adequate | 3006 | 74.8 | 74.7 |
| More than adequate | 37 | 0.9 | 1.0 |
| Missing | 1 | 0.0 | – |
| **Income sufficiency** | | | |

sample. The sociodemographic and socioeconomic characteristics of the subgroup aged 50 years and older are presented in table 3.

There was an uneven distribution of survey participants across the national EquityTool quintiles generated from the 2013 DHS. The poorest two quintiles were underrepresented (only 9% of the sample were in the poorest quintile (Q1) and 14% were in Q2) while the wealthiest three quintiles were over-represented (table 3). For all subsequent analyses, in place of these groups, we used the evenly distributed within-sample asset wealth quintiles that we generated. Very few participants placed themselves in the wealthiest category of any of the three subjective measures. For subsequent analyses we combined the two highest categories of income sufficiency because of the very small number of participants in the wealthiest group and combined the poorest two groups to leave three levels, consistent with the food adequacy measure. Over half the participants had received a non-formal (Quranic) education; this did not allow for an ordinal scale of education levels to be compared with other SEP measures and education was not included in the analyses.

### Relationship between objective and subjective SEP measures in the sample 50 years and older

For the population 50 years and older, the within-sample asset wealth quintiles showed weak positive correlation with each of the subjective food adequacy, income sufficiency and economic ladder question responses (Kendall's tau-b statistic ($\tau_b$)=0.09, 0.15 and 0.18, respectively; N=4019, 4020 and 3987, respectively; p<0.001 for all three estimates).

### Key eye health outcomes among the population 50 years and older

Among the total population 50 years and older, the age-sex weighted prevalence of blindness was 2.4% (95% CI 1.9% to 3.0%) and of any VI was 27.2% (95% CI 25.4% to 29.0%). In the same population, the age-sex weighted CSC and eCSC were 63.1% (95% CI 57.6% to 68.3%) and 29.1% (95% CI 24.2% to 34.6%), respectively at the <6/60 threshold for cataract surgery. At the lower threshold for cataract surgery of <6/12, the CSC and eCSC estimates were 33.9% (95% CI 30.5% to 37.5%) and 15.3% (95% CI 12.6% to 18.4%), respectively.

### Weighted prevalence of VI and CSC in the population 50 years and older disaggregated by SEP measures
#### Objective relative asset wealth (EquityTool)
No pattern of inequality was identified across within-sample wealth quintiles for any VI, blindness, CSC or eCSC at either threshold and there were no statistically significant differences between any of the quintiles (figures 1–3 and online supplemental appendix table S1). Consolidating within-sample quintiles to a binary comparison of asset wealth groups showed the weighted prevalence of any VI and blindness and weighted coverage (by any measure) in the poorest one or two quintiles was

**Table 3** Continued

| | Population 50 years and older (N=4020) | | |
|---|---|---|---|
| | N | Crude (%) | Weighted (%) |
| Not enough, must borrow | 922 | 22.9 | 22.6 |
| Not enough, use savings | 508 | 12.6 | 13.2 |
| Just enough | 2418 | 60.2 | 59.9 |
| Enough, save a little | 167 | 4.2 | 4.1 |
| Enough, build savings | 5 | 0.1 | 0.1 |
| Missing | 0 | 0.0 | – |
| Economic ladder question (ELQ) | | | |
| Step 1—poorest | 227 | 5.7 | 5.5 |
| Step 2 | 1240 | 30.9 | 29.5 |
| Step 3 | 1919 | 47.7 | 49.3 |
| Step 4 | 489 | 12.2 | 12.8 |
| Step 5—richest | 112 | 2.8 | 3.0 |
| Missing | 33 | 0.8 | – |

*Non-formal (Quranic) describes a system where students study mostly religious education at home for a non-specific number of years without any formal curriculum.
†Includes Madrassa and non-Madrassa systems (Madrassa describes schools where Arabic is the medium of instruction and with emphasis on Islamic education).
DHS, Demographic and Health Survey.

not significantly different to the top four or three, respectively, or to the overall weighted prevalence or coverage estimates (online supplemental appendix table S2).

### Subjective measure: economic ladder question

Almost half of participants placed themselves on the middle position of the ladder (step 3); three quarters of responses were 'clumped' together across steps 2 and 3 (table 3). There was no pattern of inequality in any VI, blindness, CSC or eCSC across the five economic ladder steps. Step 3 (the largest group) had a significantly higher prevalence of any VI compared with steps 1, 2 and 5 while there were no significant differences in blindness prevalence across the ladder (figure 1). There were few participants or cataract cases placed at either end of the scale and CIs for the estimates in these positions were wide, however, $eCSC_{<6/60}$ was significantly higher for people who placed themselves on step 2 compared with step 3 (figure 2) (see online supplemental appendix table S3).

### Subjective measure: household food adequacy

Only 1% of people reported having more than adequate household food, while 24% reported less than adequate household food (table 3). Considering the point estimates, a queuing pattern of inequality was apparent across food adequacy levels for almost all eye health outcomes, however, CIs for the estimates in the top group were very wide. Any VI, blindness, $CSC_{<6/60}$ and $eCSC_{<6/60}$ were all significantly worse among those with less than adequate food compared with those with just adequate food (figures 1 and 2) while the difference between the two was not significant for $CSC_{<6/12}$ and $eCSC_{<6/12}$ (figure 3) (see online supplemental appendix table S4).

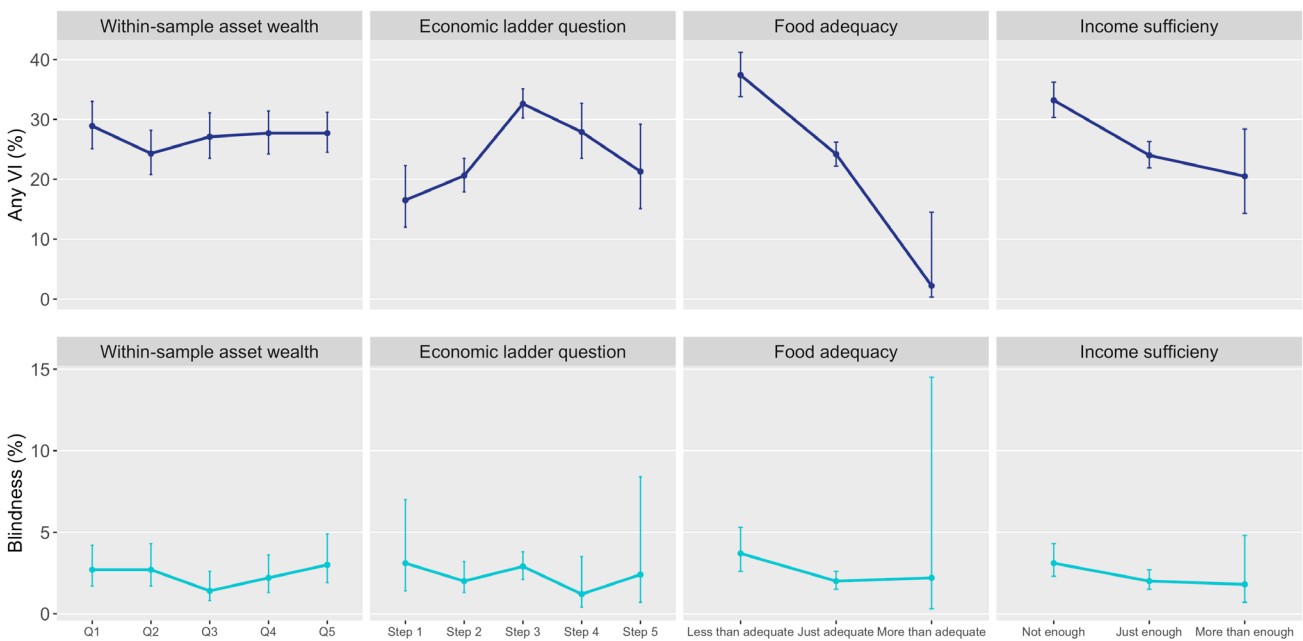

**Figure 1** Age-sex weighted prevalence of blindness and any vision impairment (VI) in the population 50 years and older by four socioeconomic position variables, The Gambia 2019.

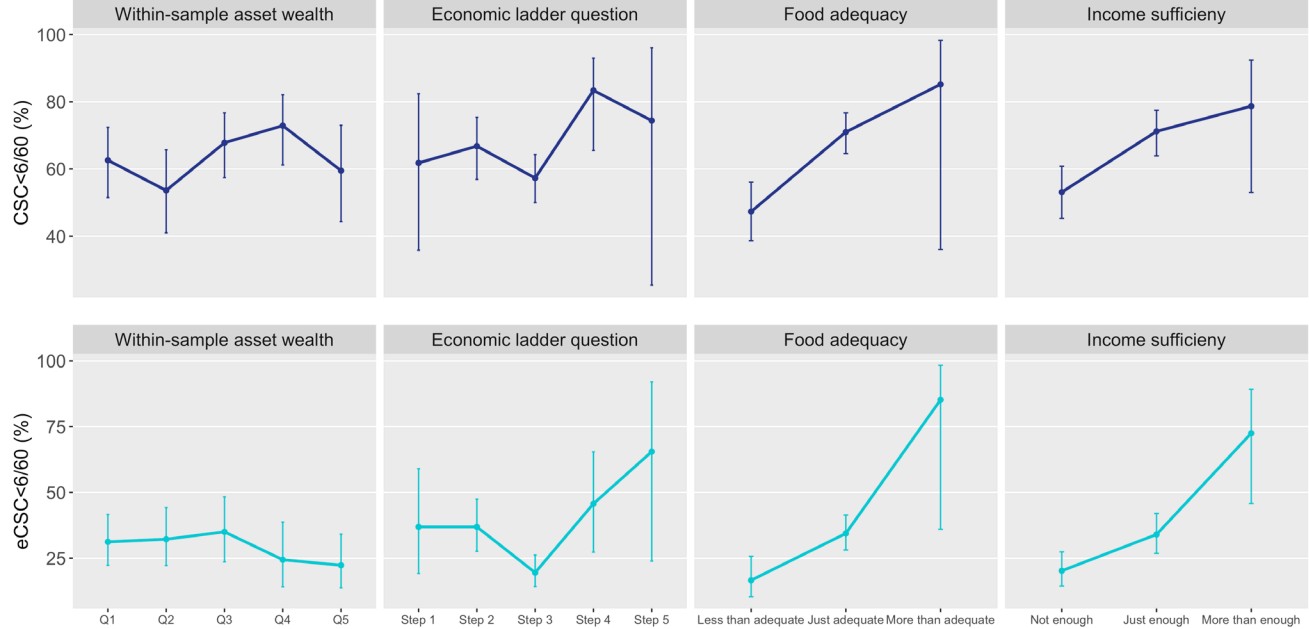

**Figure 2** Age-sex weighted cataract surgical coverage (CSC) and effective cataract surgical coverage (eCSC) (cataract surgical threshold <6/60) in the population 50 years and older by four socioeconomic position variables, The Gambia 2019.

**Subjective measure: household income sufficiency**

After consolidating the original five levels of income sufficiency to three, still only 4% of people reported 'more than enough' household income to be able to 'save a little' (level 4) or 'build savings' (level 5). Combined, 36% of participants occupied the two original 'not enough' levels of household income ('must borrow' and 'use savings') (see online supplemental appendix table S5). Considering the point estimates, a queuing pattern of inequality was seen across three levels of income sufficiency for all eye health outcomes. As for food adequacy, the low numbers of participants and cataract cases in the top group meant CIs were very wide, but there were significant differences in any VI and CSC$_{<6/60}$ between the middle and bottom groups (figures 1 and 2).

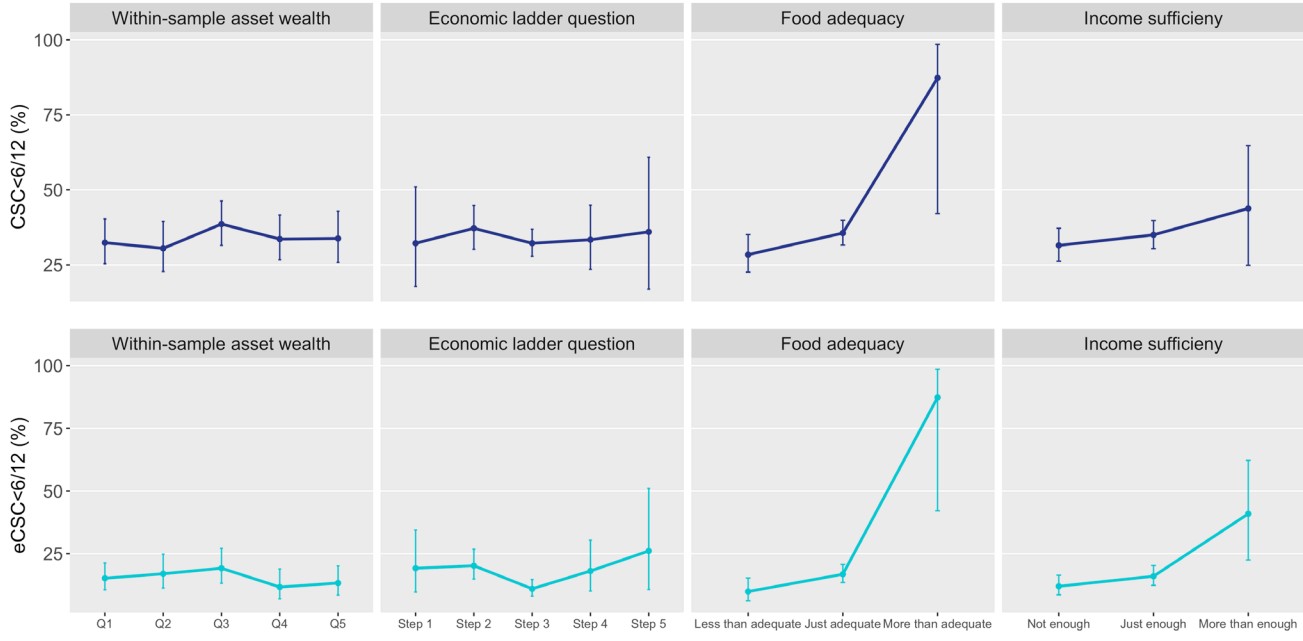

**Figure 3** Age-sex weighted cataract surgical coverage (CSC) and effective cataract surgical coverage (eCSC) (cataract surgical threshold <6/12) in the population 50 years and older by four socioeconomic position variables, The Gambia 2019.

## DISCUSSION

The 2019 Gambia National Eye Health Survey estimated key eye health outcomes in a nationally representative sample of adults aged 35 years and older and included one objective measure of asset-based relative wealth (EquityTool) and three subjective measures of SEP. We analysed data on a sub-group of 4020 adults aged 50 years and older.

The Gambia's 2015–2016 Integrated Household Survey estimated an absolute measure of poverty (poverty line) based on per capita consumption. The proportion of the population living in poverty was 48.6%, an increase of 0.5% since 2010.[29] While an absolute measure like this may be a more intuitive concept of poverty, we focused on relative SEP measures as they identify inequality in all settings (including where absolute poverty decreases with development) and are well aligned with the aim of monitoring eye health within UHC.

We found no patterns of inequality or significant differences between quintiles for any of our selected eye health outcomes by relative objective asset wealth. Similarly, a 2020 RAAB survey in rural Nigeria included an EquityTool questionnaire benchmarked to 2013 national DHS data.[30] The proportion of participants in each quintile was not reported, but there was no pattern of inequality or significant differences in blindness, severe VI or moderate VI across wealth quintiles. Asset-based measures such as EquityTool are easier to capture than income or expenditure, particularly in low/middle-income countries (LMIC), and participant responses can often be verified at interview.[8] However, certain assets may have a different relationship with SEP in different subgroups, which must be accounted for in analyses and interpretation. Even then, asset ownership does not distinguish between asset quality and assets may not reflect the short-term resource availability often required to make out-of-pocket payments for healthcare.[31] In addition, wealth indices may conflate community-level factors with household-level factors,[8] and where national thresholds are used in a relatively homogenous subnational region, as can be surveyed with RAAB, respondents may be 'clumped' in one or two quintiles.[32] From the perspective of a rapid survey methodology, multiple additional questions per individual or household may increase the time and cost of data collection to the point where it becomes a barrier to eye health planners embarking on a survey.

Howe *et al* proposed three desirable features of an SEP indicator for use in health research or evaluation in LMIC: that it is feasible in terms of cost and complexity, reliable and reproducible and that the underlying social stratification process be understood.[8] In terms of their simplicity, any of the three subjective SEP measures reviewed here would be feasible to incorporate into a rapid survey protocol and the stratification generated by each is straightforward.

The subjective economic ladder question did not identify any pattern of inequality in outcomes across the five steps. In the two most frequently selected positions (steps 2 and 3), any VI and $eCSC_{<6/60}$ were significantly better in the poorer of the two SEP positions. However, subjective SEP has been shown to be a useful predictor of health in older people elsewhere,[33] and the economic ladder question may be worth reassessing in vision and eye health surveys in other settings. We presented an image of a ladder with five steps. Participants may have been distributed differently across a nine-step or ten-step ladder (as has been used in other studies[34 35]) which may have shown a pattern of inequality not identified here.

The three-tiered subjective measures of household food adequacy and income sufficiency demonstrated that very few people in The Gambia self-identify as having more than enough of either resource. The concept of food security encompasses the need to have the physical or financial means to obtain quality food on a regular and sustained basis.[36] In LMIC, food has been shown to be households' single biggest expense,[37] and food inadequacy may be a useful tool to identify the people with fewest resources in such settings. In 2015–2016, food accounted for almost 60% of total household consumption expenditure in The Gambia.[29] A direct, subjective measure of food adequacy may be better than objective measures of adequacy, such as consumption or body mass index, given natural variations in physiology and the absence of agreed norms.[38] Historically, The Gambia has suffered from food insecurity, with levels increasing in recent years on account of climate-related flood and drought.[39] Almost a quarter of participants aged 50 years and older reported less than adequate household food and people who self-identified in this bottom group experienced worse eye health outcomes than those in the middle group. Outcomes were best in the top group but CIs for the estimates were very wide due to the very small group size. Food poverty is also prevalent in high-income countries and also associated with households' financial resources.[36] In high and some middle-income settings, the smallest group may commonly be the 'less than adequate' group rather than the 'more than adequate' seen here. In these settings, a binary comparison can be made between 'more than adequate' and 'just adequate' in place of 'just adequate' and 'less than adequate'.

Subjective income sufficiency has a complex relationship with actual income and subjective well-being and may vary over the life course.[40] In a study of people 50 years and older across 12 high-income countries it was most strongly associated with individuals' net worth and employment status, but may be over-estimated in the oldest age groups.[40] In The Gambia, just over a third of participants aged 50 years and older reported insufficient household income such that the household had to borrow money or spend savings to meet their needs. Eye health outcomes stratified by self-reported household income sufficiency followed a similar trend to food adequacy. Point estimates were all lowest in the bottom group, with any VI and $CSC_{<6/60}$ outcomes significantly worse than the middle group.

The distribution of responses to food adequacy and income sufficiency limited the extent to which any

pattern of inequality could be reliably interpreted as the measures were effectively reduced to binary 'not enough' versus 'just enough' comparisons. However, as potential eye health equity indicators, they were able to demonstrate that people who perceived themselves to have the least resources were significantly worse off for any VI prevalence, and CSC and eCSC at the <6/60 threshold. In The Gambia, eCSC was low in the overall population 50 years and older at both thresholds (eCSC$_{<6/60}$ 30%; eCSC$_{<6/12}$ 15%). Point estimates for those with insufficient income were lower (21% and 12%, respectively) and lower again for inadequate food (16% and 10%, respectively). These subjective, experience-based SEP measures may capture less obvious aspects of social status and allow research participants to be centred in the process of identifying socioeconomic inequality in each survey setting.[8] Tracking eCSC in socioeconomic groups stratified by food adequacy and/or income sufficiency may be a useful way to monitor eye health equity within countries over time and is aligned with WHO's emphasis on countries achieving a 30 percentage points increase in all population subgroups. Unlike SEP quintiles, the relative size of the self-reported groups may increase or decrease over time and, alongside changing effective coverage, this information may also be useful for national and subnational service planning.

The objective relative asset wealth metric showed weak positive association with each of the three subjective SEP measures used. Varying levels of agreement between objective and subjective SEP metrics have been reported in the literature. In eye health, a cataract VI and poverty case–control study across three LMICs found that measures of per capita expenditure, asset ownership and self-rated wealth were highly correlated.[37] A trachoma and poverty case–control study in Ethiopia found moderate agreement between a within-study asset-wealth index and self-rated wealth.[41] A 2004–2005 national household survey in Malawi found poor agreement between an objective asset-wealth index and the same subjective SEP measures we used here and the authors noted that the two approaches appeared to be measuring different concepts.[8] Individual indicators of SEP may be correlated but are rarely proxies for each other as they variously represent unique and shared aspects of the concept.[42] However, to identify a socioeconomic group at risk of—or demonstrate a socioeconomic gradient in—a poor health outcome such as VI, the choice of relative SEP indicator may not be crucial.[8]

RAAB surveys can be nationally representative but have more often been carried out at the subnational level, where participants' place of residence is often either predominantly rural or urban. We considered our sample size of around 4000 people 50 years and older too small to examine the effect of intersectionality on eye health outcomes and did not report outcomes by SEP in urban and rural populations separately. This is of relevance for rare outcomes such as blindness and cataract service coverage indicators where the number of observations in subgroups may be small. However, intersectionality

has been shown to be an important consideration in eye health,[43] and disaggregation by two equity dimensions may be feasible for some outcomes in larger RAAB surveys. Further research is required to address this question in future surveys. Another consideration for future research is the use of small-area deprivation indices such as those already employed in some high-income settings.[44] Pre-existing metrics could be applied at the cluster level and be used alongside or instead of individual level SEP data.

Our study had several limitations. Our sample was not evenly distributed across the quintiles of asset wealth generated by the EquityTool based on the 2013 DHS. Another DHS was completed in The Gambia in 2019–2020 but the subsequent EquityTool was unavailable at the time of our survey. The questions in the EquityTool elicit durable asset ownership and household characteristics. Given six years had passed between the 2013 DHS on which the EquityTool was based and our survey, we assume ongoing urbanisation and improved water and sanitation likely contributed to the under-representation of lower SEP quintiles and over-representation of higher SEP quintiles in our sample.[45] There may have been stronger associations between participant responses to the subjective SEP questions and a more contemporary or comprehensive asset wealth index. We did not assess the reliability or repeatability of the subjective SEP measures in this study, nor did we assess their acceptability in the population; however, previous research in a low-income population in the USA indicates self-reporting subjective income sufficiency may be less contentious than reporting a specific income level.[46] The wording of the questions may need to be refined in different settings to ensure they make sense to the population. These issues should all be addressed in future research. A comparison of the effects of different SEP dimensions (including those not described here) on VI status may help identify which specific socioeconomic characteristics or resources are most important for good eye health outcomes. We did not try to estimate the effects of various SEP measures on, for example, VI prevalence. Any such analysis of multiple SEP indicators requires careful causal interpretation to avoid mutual adjustment fallacies, and was beyond the aims of this study.[42]

We propose pilot testing the inclusion of two subjective SEP indicators—food adequacy and income sufficiency—in future RAAB surveys. Both SEP indicators demonstrated an inequality gradient in eye health outcomes in the population 50 years and older and may be useful for tracking progress in eCSC among population groups most in need of services. These indicators should be tested in a variety of countries with different income levels and the acceptability of each question should be reviewed in different cultural contexts.

**Author affiliations**
[1]International Centre for Eye Health, London School of Hygiene & Tropical Medicine, London, UK
[2]Sheikh Zayed Regional Eye Care Centre, Banjul, Gambia

³MRC Unit The Gambia at LSHTM, Banjul, Gambia
⁴Directorate of Health Promotion and Education, Ministry of Health, Kotu, Gambia
⁵Gambia Bureau of Statistics, Ministry of Finance, Kanifing, Gambia
⁶Directorate of Planning and Information, Ministry of Health, Kotu, Gambia
⁷Regional Directorate of Health Services, West Coast Health Region, Ministry of Health, Kanifing, Gambia
⁸National Institute for Health Research Biomedical Research Centre for Ophthalmology, Moorfields Eye Hospital NHS Foundation Trust, London, UK
⁹School of Optometry and Vision Science, The University of Auckland, Auckland, New Zealand

**Acknowledgements** JR's position at the University of Auckland is funded by the Buchanan Charitable Foundation, New Zealand.

**Contributors** JR conceptualised the research question. IMcCormick, MJK, AH, SIO, MJ, AB, IMactaggart, MJB and JR contributed to the design and conduct of the survey. IMcCormick, MJK and IMactaggart analysed the data. OB, NMBS, GJ, MN, AS and LA contributed to interpreting the results. IMcCormick drafted the original manuscript and accepts full responsibility for the work as guarantor. All coauthors reviewed and edited the manuscript.

**Funding** This research was funded by The Queen Elizabeth Diamond Jubilee Trust [Grant number TG002]. MJB is supported by the Wellcome Trust [207472/Z/17/Z].

**Competing interests** None declared.

**Patient and public involvement** Patients and/or the public were not involved in the design, or conduct, or reporting, or dissemination plans of this research.

**Patient consent for publication** Not applicable.

**Ethics approval** Ethical approval for the study was granted by The Gambia Government/Medical Research Council Joint Ethics and Scientific Coordinating Committee (Ref SCC 1635) and the London School of Hygiene & Tropical Medicine Research Ethics Committee (Ref 16172). All participants provided written informed consent to participate in the 2019 Gambia National Eye Health Survey.

**Provenance and peer review** Not commissioned; externally peer reviewed.

**Data availability statement** Data are available upon reasonable request.

**ORCID iDs**
Ian McCormick http://orcid.org/0000-0002-7360-3844
Min J Kim http://orcid.org/0000-0003-3561-9308
Abba Hydara http://orcid.org/0000-0003-1426-3808
Segun I Olaniyan http://orcid.org/0000-0002-2006-6897
Modou Jobe http://orcid.org/0000-0001-5309-5611
Andrew Bastawrous http://orcid.org/0000-0001-8179-556X
Luke Allen http://orcid.org/0000-0003-2750-3575
Islay Mactaggart http://orcid.org/0000-0001-6287-0384
Matthew J Burton http://orcid.org/0000-0003-1872-9169
Jacqueline Ramke http://orcid.org/0000-0002-5764-1306

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
