## [Reviewer comments · BMJ Open]

ARTICLE DETAILS

TITLE (PROVISIONAL)	Socioeconomic position and eye health outcomes: identifying inequality in rapid population-based surveys
AUTHORS	McCormick, Ian; Kim, Min; Hydera, Abba; Olaniyan, Segun I; Jobe, Modou; Badjie, Omar; Sanyang, Nyakassi MB; Jarju, Gibril; Njai, Modou; Sankareh, Alhagie; Bastawrous, Andrew; Allen, Luke; Mactaggart, Islay; Burton, Matthew J; Ramke, Jacqueline

VERSION 1 – REVIEW

REVIEWER	Edward Saxby Princess Alexandra Eye Pavilion
REVIEW RETURNED	08-Nov-2022

GENERAL COMMENTS	Well written paper of significant scientific interest
---

REVIEWER	Hanne-Mari Schiøtz Thorud University of South-Eastern Norway
REVIEW RETURNED	11-Nov-2022

GENERAL COMMENTS	Introduction I suggest you include rates of the most frequent causes of vision impairment in Gambia among adults aged 50 years and older, relative to other countries and globally. A description of eye care services in Gambia would also be useful for the later interpretation of the results. Methods 'Data collection and categorisation': I suggest you improve the clarity of study characteristics, including which age group and outcomes of The 2019 Gambia National Eye Health Survey that were included in this study. '2013 National Census': Include a reference. I suggest you describe how eye outcome measures were selected from The Gambia National Eye Health Survey 2019 for this study. Uncorrected refractive error is a leading cause of moderate or severe vision impairment globally. Effective refractive error coverage (eREC) in low- and middle-income regions is reported to be very low among adults aged 50 years and older. Why was not eREC included in this study? The frequency of unaddressed near vision impairment is reported to be very high in western, eastern and central Sub-Saharan Africa. Why was distance vision impairment, but not near vision impairment, included in this study? Distance visual acuity (VA): Describe the measurements in detail or refer to the 'Eye health assessment' section in The 2019 Gambia National Eye Health Survey. 'In reporting selected eye health outcomes, we restricted our population of interest to people 50 years and older as this age group
--

	is directly relevant to the RAAB survey methodology.': This sentence should be moved to the 'Data collection and categorisation' section. 'Missing data' – 'forms with persisting missing critical data were deleted from the final sample.': Please add details; what includes 'persisting missing critical data' and how many forms were deleted? 'Missing data' – 'EquityTool questions were imputed for 398 participants (4.3%) in the final dataset.': These numbers presumably describe the whole dataset for the The 2019 Gambia National Eye Health Survey (9188). What were the numbers for the sub-dataset in this study (4020) regarding EquityTool? Results 'There were 9,188 participants examined out of 11,027 enumerated (response rate 83%).': In 'The Gambia National Eye Health Survey 2019: survey protocol' the number is 11,127. What is correct? Table 1: How was 'Weighted %' calculated? 'National asset wealth quintiles (2013 DHS EquityTool)' – is age the only difference between these numbers and the numbers in Table 2 in In 'The Gambia National Eye Health Survey 2019: survey protocol'? How was 'Within-sample asset wealth quintiles' computed? Please add details to improve clarity. Discussion I suggest you also discuss the results in relation to eye care services and rates of vision impairment in Gambia.
--	---

REVIEWER	Denekew Anley Debre Tabor University, Public Health
REVIEW RETURNED	17-Dec-2022

GENERAL COMMENTS	It is good work. However, I have concerns like; 1) in the objective part of the abstract section you wrote something else other than the objective of the study. Avoid writing introduction while you are supposed to write the objective of the study in a measurable way using action verbs. 2) You have written about the association of your wealth related variables with the visual impairment outcomes. However, I couldn't find the regression analysis table. Would you show us , please?
--

VERSION 1 – AUTHOR RESPONSE

Reviewer: 1
Dr. Edward Saxby, Princess Alexandra Eye Pavilion

Comments to the Author:
Well written paper of significant scientific interest

Thank you for your time and positive feedback.

Reviewer: 2
Dr. Hanne-Mari Schiøtz Thorud, University of South-Eastern Norway

Comments to the Author:

Introduction

I suggest you include rates of the most frequent causes of vision impairment in Gambia among adults aged 50 years and older, relative to other countries and globally. A description of eye care services in Gambia would also be useful for the later interpretation of the results.

Thank you for your review and detailed comments. Since this paper was drafted, the main epidemiological findings of the 2019 Gambia National Eye Health Survey have been published

in a separate article. We have now included a reference to the following publication in the introduction:

Hydara A, Mactaggart I, Bell SJ, et al. Prevalence of blindness and distance vision impairment in the Gambia across three decades of eye health programming. *British Journal of Ophthalmology* Published Online First: 23 December 2021. [https://doi: 10.1136/bjophthalmol-2021-320008](https://doi.org/10.1136/bjophthalmol-2021-320008)

In the introduction we aimed not to duplicate information on the status of eye health in The Gambia published in the article above, but to emphasise the methodological question under consideration. In the interest of keeping the word count at an acceptable level, we believe the background on measurement of SEP and the requirements of rapid surveys provides the most important context for this particular research question.

Methods

'Data collection and categorisation': I suggest you improve the clarity of study characteristics, including which age group and outcomes of The 2019 Gambia National Eye Health Survey that were included in this study.

Thank you. We agree some additional justification for our outcome selection will enhance the Methods and have included it in the 'Eye health outcomes' section. We have added a sentence to explain why the eye health outcomes reported here were chosen from among many possible outcomes: *"We selected WHO priority national eye health indicators as outcomes: vision impairment prevalence and effective cataract surgical coverage."* We added a reference to the WHO's Eye Care Indicator Menu.

We included an explanation of the age group used in this analysis in the 'Data analysis' paragraph of the Methods of the original submission: *"In reporting selected eye health outcomes, we restricted our population of interest to people 50 years and older as this age group is directly relevant to the RAAB survey methodology."*

'2013 National Census': Include a reference.

Thank you. We have added a reference for the Census report.

I suggest you describe how eye outcome measures were selected from The Gambia National Eye Health Survey 2019 for this study. Uncorrected refractive error is a leading cause of moderate or severe vision impairment globally. Effective refractive error coverage (eREC) in low- and middle-income regions is reported to be very low among adults aged 50 years and older. Why was not eREC included in this study? The frequency of unaddressed near vision impairment is reported to be very high in western, eastern and central Sub-Saharan Africa. Why was distance vision impairment, but not near vision impairment, included in this study?

Please see our response above - we have added a sentence to explain the rationale for selecting vision impairment and eCSC/CSC: *"We selected WHO priority national eye health indicators as outcomes: vision impairment prevalence and effective cataract surgical coverage."*

While we agree eREC is an important outcome, the aim of this paper is not to communicate survey outcomes. eREC has been described in the following publication (and was indeed found to be very low):

Boggs D, Hydara A, Faal Y, et al. Estimating Need for Glasses and Hearing Aids in The Gambia: Results from a National Survey and Comparison of Clinical Impairment and Self-Report Assessment Approaches. *Int. J. Environ. Res. Public Health* 2021, 18, 6302. <https://doi.org/10.3390/ijerph18126302>

For the purposes of our analysis, any exploration of eREC by SEP subgroups would have been limited by very small numbers.

Distance visual acuity (VA): Describe the measurements in detail or refer to the 'Eye health assessment' section in The 2019 Gambia National Eye Health Survey.

We have added the clause “more information is available in the Eye Health Assessment section of the study protocol” with the relevant reference.

'In reporting selected eye health outcomes, we restricted our population of interest to people 50 years and older as this age group is directly relevant to the RAAB survey methodology.': This sentence should be moved to the 'Data collection and categorisation' section.

Thank you for this suggestion. We have not moved this text to the 'Data collection and categorisation' section, as it refers to an analysis choice we made for this particular paper rather than the data collection or categorisation of the underlying survey. Therefore, we believe it is best placed to remain in the 'Data analysis' section, though on reflecting on your comment we moved it to the top of this subheading.

'Missing data' – 'forms with persisting missing critical data were deleted from the final sample.': Please add details; what includes 'persisting missing critical data' and how many forms were deleted?

We have added this additional detail to the Methods section along with the number of forms deleted: “In August 2020, follow-up data collection was completed by telephone to address some missing EquityTool data and following this, 600 forms with persisting missing critical data (one or more entire module [e.g., optometry, ophthalmology, mental health], EquityTool data or visual acuity data) were deleted from the final sample.”

'Missing data' – 'EquityTool questions were imputed for 398 participants (4.3%) in the final dataset.': These numbers presumably describe the whole dataset for the The 2019 Gambia National Eye Health Survey (9188). What were the numbers for the sub-dataset in this study (4020) regarding EquityTool?

We have included the count (and percentage) of participants in the sub-dataset with imputed EquityTool responses. “EquityTool questions were imputed for 398 participants (4.3%) in the final dataset (179 [4.5%] participants aged 50 years and older), whereby missing values in a cluster were replaced with the most frequent, non-missing value from within the cluster.”

Results

'There were 9,188 participants examined out of 11,027 enumerated (response rate 83%).': In 'The Gambia National Eye Health Survey 2019: survey protocol' the number is 11,127. What is correct?

The correct value is 11,027 as reported here.

Table 1: How was 'Weighted %' calculated?

Thank you. We have added a sentence to the 'Data analysis' section of the Methods: “The weighted proportions of participants by sociodemographic and socioeconomic position characteristics were calculated according to the age-sex distribution of The Gambia 2013 Census using the svy command.”

'National asset wealth quintiles (2013 DHS EquityTool)' – is age the only difference between these numbers and the numbers in Table 2 in 'The Gambia National Eye Health Survey 2019: survey protocol'?

Yes.

How was 'Within-sample asset wealth quintiles' computed? Please add details to improve clarity.

Thank you. We have clarified the approach used by adding the following sentence to the 'Data analysis' section of the Methods: *"We assigned participants aged 50 years and older to within-sample wealth quintiles using the xtile command."*

Discussion

I suggest you also discuss the results in relation to eye care services and rates of vision impairment in Gambia.

The aim of our analysis is to demonstrate the utility or otherwise of subjective SEP measurements to demonstrate a socioeconomic gradient or inequality in key eye health outcomes in the population 50 years and older. We did not aim to describe vision impairment and the state of eye care services in The Gambia as this has been published elsewhere - we trust the interested reader will be able to engage with that information via the new reference added to the Introduction. Other publications based on the 2019 survey reporting the epidemiology of specific eye conditions in The Gambia are still in preparation.

Reviewer: 3

Mr. Deneke Anley, Debre Tabor University

Comments to the Author:

It is good work. However, I have concerns like;

1) in the objective part of the abstract section you wrote something else other than the objective of the study. Avoid writing introduction while you are supposed to write the objective of the study in a measurable way using action verbs.

Thank you for taking the time to review our paper. To introduce the stated objective of the study in the abstract ("to assess whether each of four SEP measures identified inequality – an underserved group or socioeconomic gradient – in key eye health outcomes.") we included two introductory sentences. If the editorial team are happy with this approach, we would prefer to leave them in place to aid reader comprehension.

2) You have written about the association of your wealth related variables with the visual impairment outcomes. However, I couldn't find the regression analysis table. Would you show us , please?

We did not describe an association between SEP variables and eye health outcomes in this study and did not do a regression analysis. In the discussion we noted that *"We did not try to estimate the effects of various SEP measures on e.g., vision impairment prevalence. Any such analysis of multiple SEP indicators requires careful causal interpretation to avoid mutual adjustment fallacies, and was beyond the aims of this study."*

VERSION 2 – REVIEW

REVIEWER	Hanne-Mari Schiøtz Thorud University of South-Eastern Norway
REVIEW RETURNED	25-Jan-2023
GENERAL COMMENTS	Thank you for an interesting and important paper.
REVIEWER	Deneke Anley Debre Tabor University, Public Health
REVIEW RETURNED	26-Jan-2023
GENERAL COMMENTS	Thank you for this work!!